# Significance of the Glasgow Prognostic Score in Predicting the Postoperative Outcome of Patients with Stage III Gastric Cancer

**DOI:** 10.3390/jcm8091448

**Published:** 2019-09-12

**Authors:** Shun-Wen Hsueh, Keng-Hao Liu, Chia-Yen Hung, Yung-Chia Kuo, Chun-Yi Tsai, Jun-Te Hsu, Yu-Shin Hung, Ngan-Ming Tsang, Wen-Chi Chou

**Affiliations:** 1Department of Hematology-Oncology, Chang Gung Memorial Hospital at Keelung, Keelung 204, Taiwan; 2College of Medicine, Chang Gung University, Taoyaun 333, Taiwan; 3Department of Surgery, Chang Gung Memorial Hospital at Linkou, Taoyuan 333, Taiwan; 4Department of Hematology-Oncology, Chang Gung Memorial Hospital at Linkou, Taoyuan 333, Taiwan; 5Department of Hematology-Oncology, Mackay General Hospital, Taipei 104, Taiwan; 6Department of Radiation Oncology, Chang Gung Memorial Hospital at Linkou, Taoyuan 333, Taiwan

**Keywords:** gastric cancer, Glasgow prognostic score, gastrectomy, complication, survival

## Abstract

This study aimed at investigating the ability of a preoperative Glasgow prognostic score (GPS) to predict postoperative complications and survival outcomes in patients with stage III gastric cancer undergoing D2 gastrectomy. We retrospectively reviewed data from 272 such patients, treated between 2010 and 2016, at a Taiwanese medical center. The patients were categorized according to their GPS. In total, 36.8%, 48.5%, and 14.7% of the patients were assigned to groups with a GPS of 0, 1, and 2, respectively. Overall surgical complication rates in these groups were 30%, 45.5%, and 52.5% (*p* = 0.016); postoperative intensive care unit admission rates were 10%, 14.4%, and 22.5% (*p* = 0.15); postoperative 30-day re-admission rates were 6%, 15.2%, and 20% (*p* = 0.034); and the in-hospital mortality rates were 1.0%, 1.5%, and 10.0%, respectively (*p* = 0.006). The median survival times of the patients were 42.9 months (95% confidence interval [CI], 29.1–56.6), 22.6 months (95% CI, 19.3–25.8), and 16.6 months (95% CI, 7.8–25.4), respectively (*p*< 0.001). A significant correlation was observed between the preoperative GPS, short-term postoperative complications, and long-term survival outcomes in patients with gastric cancer undergoing D2 gastrectomy. These findings recommend the usage of the GPS as a predictive and prognostic factor in patients with gastric cancer considering surgical resection.

## 1. Introduction

As the fourth most commonly diagnosed malignancy in the world, gastric cancer is a fatal disease that accounted for 5.7% of all newly diagnosed cancer cases and 8.2% of cancer-related deaths in 2018 [1]. In Taiwan, gastric cancer is the ninth most common cancer, which accounted for 3.5% of all newly diagnosed cancer cases and 4.9% of cancer-related deaths in 2016 [2]. Both in the US and in Taiwan, around one-fourth of the patients diagnosed with gastric cancer presented with a locally advanced stage during diagnosis [2,3]. Although radical surgery is the only curative treatment modality for locally advanced gastric cancer, the postoperative outcome remains substandard, with nearly half of the patients experiencing tumor recurrences within a short time [4,5]. Furthermore, short-term postsurgical complications are considerable issues, since up to 15% of the patients experience postsurgical complications, as per a recent study [6]. In addition to postoperative risks and high tumor recurrence rates in patients with locally advanced diseases [7], suboptimal survival outcomes were also observed due to chances of complications in radical surgery [8]. The Taiwan National Cancer Register data showed that the 5-year survival rate ranged from 13.1% to 43.8% in patients with locally advanced gastric cancer who received radical surgery [2]. As radical surgery is the only curative strategy for gastric cancer, a predictive tool identifying high risk patients prone to short-term operation complications and poor long-term survival would assist clinicians and patients in adopting appropriate treatment strategies.

In current clinical practice, inflammatory-based methods have been developed to predict the postoperative morbidity and mortality of various cancers after radical surgeries. Among the different inflammation-based cancer-prognostic markers, Glasgow prognostic score (GPS) has been extensively investigated for use in various cancers [9,10,11]. It is defined as the combination of serum C-reactive protein (CRP) and albumin levels, which are indicators of systematic inflammatory response and nutritional status, respectively. Previous studies have reported an elevated GPS as a negative prognosticator in elderly patients or patients with stage IV gastric cancer receiving palliative gastrectomy [12,13,14,15,16,17]. Limited studies have used the GPS for postoperative prognostication of patients with gastric cancer [18]. As for the regional disease, stage III patients have the highest risk of tumor recurrence and surgical morbidity [7], which often leads to greater clinical dilemmas. We hope that by focusing a study on stage III gastric cancer, the GPS can provide surgeons and patients with a more informed clinical direction. The aim of this retrospective study was to evaluate the significance of the GPS as a predictive and prognostic factor of survival in patients with stage III gastric cancer after receiving curative-intent gastrectomy.

## 2. Materials and Methods

### 2.1. Patient Selection and Treatment

A total of 509 consecutive patients with stage III gastric cancer who underwent radical gastrectomy and D2 lymph node dissection surgery between 2007 and 2014 at Linkou Chang Gung Memorial Hospital (CGMH) were retrospectively analyzed in this study. Tumor staging was defined according to the 7th edition American Joint Committee on Cancer (AJCC) staging system, after pathological examination. Patients with CRP and albumin levels recorded within 7 days before radical surgery were further selected. Patients who underwent palliative surgery, received neoadjuvant chemotherapy, or radiotherapy before surgery, or had concurrent active malignancy, were excluded. In total, 272 patients were included in the final analysis. Patients were further categorized into 3 groups, stratified by their GPS, for survival and postoperative complication analysis.

The decision as to whether a patient should undergo a total or subtotal gastrectomy was made by the surgeon, based on the tumor location and resection margin. All of the patients were advised to begin adjuvant chemotherapy within 6 weeks of their gastrectomy. The chemotherapeutic dosage and the treatment schedule were determined by the primary care physician, as described in our previous study [19]. The final decision to proceed with adjuvant chemotherapy was left to the patient’s discretion. This study was approved by the Institutional Review Boards of all the CGMH branches, and was performed in compliance with the Helsinki Declaration (1996).

### 2.2. Data Collection and Follow-Up

The demographic data of patients, including age, sex, body mass index, Eastern Cooperative Oncology Group (ECOG) performance status, pre-existing comorbidities assessed by a modified Charlson comorbidity index (CCI) [20], anatomic location of the primary cancer, serum carcinoembryonic antigen (CEA) levels, carbohydrate antigen 19-9 (CA 19-9) levels, histological differentiation, pathological T- and N- classification, gastrectomy method, and use of adjuvant chemotherapy, were recorded by the primary care physician using a prospectively formulated electronic data form derived from our previous study. All postoperative events were recorded by a retrospective chart review. Postsurgical complications were defined using the Clavien−Dindo Classification [21], which classified complications of grade 3 or higher as severe surgical complications. The overall survival time was defined as the time from the date of surgery to the date of death by any cause, or the last objective information registered in the medical chart. All patients were followed up until death or until 30 June 2016. 

### 2.3. Glasgow Prognostic Score

A GPS score, calculated from the CRP and albumin levels obtained prior to surgery, was assigned to each patient. Patients with both an elevated CRP level (>10.0 mg/dL) and hypoalbuminemia (<3.5 g/dL) were assigned a score of 2. Patients with only one of the two abnormal values were assigned a score of 1. Patients who had both CRP and albumin levels within normal ranges were assigned a score of 0. 

### 2.4. Statistical Analysis 

The demographic data were analyzed as median with 95% confidence intervals (CI). Differences between the different GPS groups were determined using the Pearson chi-squared (χ2) test. The log-rank test was used to perform univariate and multivariate analyses of all clinical factors, based on overall survival (OS). All variables in the univariate analysis with *p* values of < 0.05 were further analyzed using multivariate analysis. Survival time was analyzed using the Kaplan–Meier method. Study groups, stratified according to their GPS score, were compared using the log-rank test, and hazard ratios (HRs) were obtained using a stratified Cox proportional hazards model. SPSS 17.0 software (SPSS Inc., Chicago, IL, USA) was used for statistical analysis. All statistical assessments were 2 sided, and a *p* value of <0.05 was considered statistically significant.

## 3. Results

The demographic data of patients are presented in Table 1. Of the 272 included patients (176 men; 96 women), 100 (36.8%), 132 (48.5%), and 40 patients (14.7%) were assigned a GPS of 0, 1, and 2, respectively. Patients with a GPS of 2 had a higher prevalence of poorer ECOG performance, a body weight loss of ≥ 5%, an advanced T-classification, and an advanced AJCC tumor stage. Distributions based on age, sex, Charlson comorbidity index, CEA, CA19-9, N-classification, lymphatic invasion, operation method, resection margin, and the use of adjuvant chemotherapy did not have significant statistical in-group difference.

The short-term postoperative complications which occurred after gastrectomy are presented in Figure 1. For patients with a GPS of 0, 1, and 2, the overall surgical complication rates were 30%, 45.5%, and 52.5%, respectively (*p* = 0.016); the severe surgical complication rates were 8.0%, 23.5%, and 22.5%, respectively (*p* = 0.006); the postoperative intensive care unit (ICU) admission rates were 10%, 14.4%, and 22.5%, respectively (*p* = 0.15); the postoperative 30-day re-admission rates were 6%, 15.2%, and 20%, respectively (*p* = 0.034); and the in-hospital mortality rates were 1.0%, 1.5%, and 10.0%, respectively (*p* = 0.006). 

In total, 188 patients (69.1%) died by the end of the study period and the median overall survival time was 27.3 months (range, 0.7–92.5 months). The median survival times of patients with a GPS of 0, 1, and 2, were 42.9 months (95% CI, 29.1–56.6), 22.6 months (95% CI, 19.3–25.8), and 16.6 months (95% CI, 7.75–25.4), respectively (Figure 2). When patients with a GPS of 1 and 2 were compared with those with a GPS of 0, their HRs were 2.52 (95% CI, 1.800–3.521, *p*< 0.01) and 2.26 (95% CI, 1.449–3.532, *p* < 0.001) respectively. There was no significant survival difference between the GPS 1 and GPS 2 groups (*p* = 0.92).

Univariate analysis of factors predictive of overall survival is shown in Table 2. Nine factors were found to be significantly associated with overall survival: ECOG performance, Charlson comorbidity index, AJCC stage, body weight loss, vascular invasion, resection margin, operation method, use of adjuvant chemotherapy, and the GPS. Multivariate analysis revealed a significant association between postoperative mortality and ECOG performance, AJCC stage, body weight loss, use of adjuvant chemotherapy, and the GPS. The adjusted HRs were 1.97 (95% CI, 1.36–2.86; *p* < 0.001) when the GPS 1 and GPS 0 groups were compared, and 1.57 (95% CI, 0.96–2.57; *p* = 0.071) when the GPS 2 and GPS 0 groups were compared.

## 4. Discussion

This retrospective study analyzed data from 272 patients with postoperative stage III gastric cancer, treated at a high-volume medical center in Taiwan. The results demonstrated that an elevated GPS, which is an inflammation-based prognostic score, is associated with higher short-term postoperative risks and shorter long-term survival in patients with stage III gastric cancer. Our multivariate analysis revealed that while the common prognostic factors of malignancy, including ECOG performance, AJCC stage, body weight loss, and use of adjuvant chemotherapy, are consistently valuable prognostic factors, the GPS is also an independent marker of poor prognosis in advanced gastric cancer patients. As the GPS also has the advantage of easy procurement prior to surgical intervention, it may be used as part of the routine evaluation during advanced gastric cancer treatment planning. 

In various studies [22,23], CRP levels have been used in detecting the systemic inflammation involved in malignancy, as elevated serum CRP levels may be associated with tumor size, vascular invasion, lymph node metastasis, distant metastasis, and tumor recurrence. Hypoalbuminemia is often used as a malnutrition and cachexia index, leading to poor prognosis in various cancers [24,25,26]. Since both systemic inflammation and malnutrition contribute to worse survival outcomes in malignancy, the GPS, which incorporates elevation of CRP levels and hypoalbuminemia, is a possible independent prognostic indicator for worse prognosis. In the literature [9], the GPS has been compared with other biochemical parameters, and its prognostic value has been shown. Although the prognostic significance of the GPS in gastric cancer has previously been reported [16,27], short-term and long-term postoperative outcomes in advanced gastric cancer have not been fully examined [18]. In our study focused on stage III gastric cancer, we noted that the GPS correlated significantly with ECOG performance, body weight loss, T stage, and AJCC classification, suggesting that an elevated GPS correlates with a more advanced disease.

As our study solely enrolled patients with operable stage III gastric cancer undergoing radical surgery with curative intents, we were able to evaluate patients’ short-term postsurgical complications, including overall surgical complication, severe surgical complication, postoperative ICU admission, 30-day re-admission, and in-hospital mortality. Of all the postoperative aspects assessed, only postoperative ICU admission rates showed no statistical significance (*p* = 0.153). This may be partially attributed to physicians’ subjective decisions to admit patients to intensive care. Significant differences among patients with a different GPS were observed in other postoperative complications. Notably, patients with a GPS of 2 had an appalling in-hospital mortality rate of 10%, while for patients with a GPS of 0 and 1, the in-hospital mortality rates were 1.0% and 1.5% respectively (*p* = 0.006), indicating ostensible poor prognosis from surgery in patients with a GPS of 2. Patients with a GPS of 0 had a surgical complication rate of 8%, as opposed to 23.5% and 22.5% (*p* = 0.006) in patients with a GPS of 1 and 2 respectively, revealing that patients with a GPS of 0 are more likely to endure radical surgery. Therefore, the ability of the GPS to identify patients with greater surgical risks has been made evident, and it could contribute to patients’ informed decision making prior to surgical intervention. 

The GPS provides a preoperative means to predict short-term postoperative outcomes. Since advanced gastric cancer patients could be identified with different ranges of surgical risks according to their GPS, the GPS could enable surgeons to adjust the aggressiveness of their surgical approach. Surgeons should also pay attention to perioperative care for patients with an elevated GPS. Since patients with a GPS of 2 presented with high surgical risks and prominent rates of surgical mortality, initial curative surgery may be discouraged in such patients. Other options, such as neoadjuvant treatment or a palliative approach, should be presented to these patients. However, evidence regarding the risks and benefits of neoadjuvant treatment in patients with a GPS of 2 has not been made available.

The Kaplan–Meier analysis demonstrated significant differences among patients with a GPS of 0 and those with a GPS of 1 and 2, respectively. However, differences in survival between the GPS 1 and the GPS 2 groups have not been observed (Figure 2). As shown in the plot, the GPS 2 group had a distinctly poorer prognosis than the GPS 1 group at any time prior to two years after surgery. While the GPS 2 group had the least number of patients, the number of patients in this group declined from 40 to 12, due to deaths during the 2-year follow-up period after surgery. The small study number that remained rendered the follow-up study particularly susceptible to bias. We believe that in a larger study population, the GPS 1 group’s survival advantage over the GPS 2 group will be distinct and consistent any time after the surgery. Even though there is no evidence of survival differences between the GPS 1 and the GPS 2 groups, our study has revealed that patients with a GPS of 0 have a perceptible survival advantage when compared with patients with a GPS of 1 or 2. 

Among the study cohort, 70.2% of the patients received adjuvant chemotherapy. Unequal distribution is noted, as the GPS 2 group has a lower probability to receive adjuvant chemotherapy (57.5%) than those in the GPS 0 group (74.0%). However, the postoperative short-term complications were not influenced by the administration of adjuvant chemotherapy, since these events were recorded prior to the initiation of adjuvant chemotherapy. Besides the administration of adjuvant chemotherapy, different ECOG performances might also have an influence on long-term survival. To demonstrate the significance of the GPS despite the unequal distributions of ECOG performance and adjuvant chemotherapy administration, multivariate analysis was performed, and revealed that the GPS has a significant effect on long-term survival after adjusting for these confounding factors. 

There are several limitations to this study. First, this is a retrospective study with a single-center design. Second, patient selection bias may be present since patients with operable gastric cancer who opted out of surgery, and patients in whom metastasis was discovered perioperatively, were excluded. Third, while the use of adjuvant chemotherapy was analyzed within the study, the choice of regimen was not included in the evaluation of the GPS. Large-scale future studies are necessary to prove the survival difference between the GPS 1 and GPS 2 groups. Furthermore, as a GPS of 2 is associated with a very poor short-term postoperative survival in this study, further studies may focus on whether these patients would benefit from neoadjuvant treatment. As neoadjuvant chemotherapy treatment has become one of the standard treatments for locally advanced gastric cancer during the past few years [28,29], future studies may focus on the predictive value of the GPS with patients who receive neoadjuvant chemotherapy, and the effect of the change of the GPS before and after neoadjuvant chemotherapy on the postoperative outcome.

## 5. Conclusions

Our study demonstrated that the GPS, a simple and effective scoring method to recognize a patient’s systemic inflammation and nutritional status, has a short-term postoperative and long-term survival predictive value in patients with stage III gastric cancer. Even prior to surgical intervention, the GPS may be used with other clinical considerations to optimize the treatment plan for advanced gastric cancer.

## Figures and Tables

**Figure 1 jcm-08-01448-f001:**
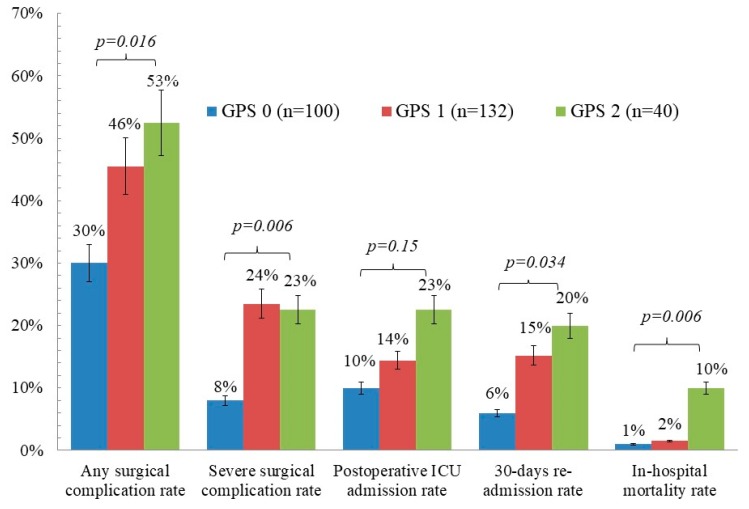
Prevalence rate of short-term postoperative complications according to Glasgow prognostic score (GPS). ICU, intensive care unit.

**Figure 2 jcm-08-01448-f002:**
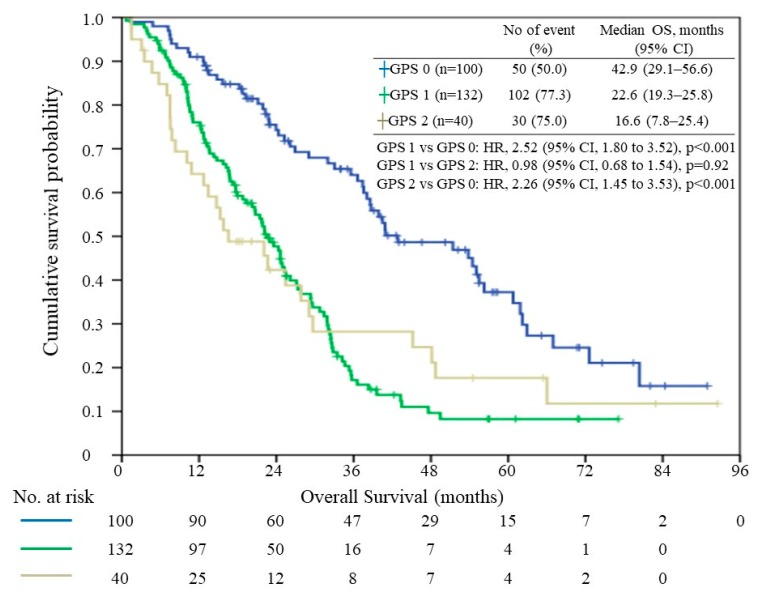
Relationship between the Glasgow prognostic score (GPS) and overall survival (OS). HR, hazard ratio; CI, confidence interval.

**Table 1 jcm-08-01448-t001:** Clinicopathological characteristics of patients grouped according to Glasgow prognostic score (GPS).

Characteristic	GPS = 0 (n = 100), N (%)	GPS = 1 (n = 132), N (%)	GPS = 2 (n = 40), N (%)	*p* Value
Median age, year (range)	63 (30–97)	64 (26–90)	70 (41–88)	0.35
Male sex	56 (56.0)	91 (68.9)	29 (72.5)	0.07
ECOG PS				<0.001
0 or 1	88 (88.0)	105 (79.5)	23 (57.5)	
2	10 (10.0)	24 (18.2)	10 (25.0)	
3	2 (2.0)	3 (2.3)	7 (17.5)	
CCI				0.11
0	50 (50.0)	59 (44.7)	11 (27.5)	
1	33 (33.0)	49 (37.1)	15 (37.5)	
2	13 (13.0)	14 (10.6)	8 (20.0)	
>2	4 (4.0)	10 (7.6)	6 (15.0)	
CEA, ng/dL				0.87
<5	85 (85.0)	111 (84.1)	35 (87.5)	
≥5	15 (15.0)	21 (15.9)	5 (12.5)	
CA19-9, ng/dL				0.54
≤37	85 (85.0)	107 (81.1)	31 (77.5)	
>37	15 (15.0)	25 (18.9)	9 (22.5)	
AJCC tumor stage				0.005
IIIA	30 (30.0)	25 (18.9)	6 (15.0)	
IIIB	42 (42.0)	39 (29.5)	17 (42.5)	
IIIC	28 (28.0)	68 (51.5)	17 (42.5)	
Operation method				0.11
Total gastrectomy	36 (36.0)	59 (44.7)	11 (27.5)	
Subtotal gastrectomy	64 (64.0)	73 (55.3)	29 (72.5)	
Resection Margin				0.038
Positive	7 (7.0)	23 (17.4)	8 (20.0)	
Negative	93 (93.0)	109 (82.6)	32 (80.0)	
Adjuvant Chemotherapy				0.15
Yes	74 (74.0)	94 (71.2)	23 (57.5)	
No	26 (26.0)	38 (28.2)	17 (42.5)	

GPS, Glasgow prognostic score; ECOG PS, Eastern cooperative oncology group performance status; CCI, Charlson comorbidity index; CEA, carcinoembryonic antigen; CA19-9, carbohydrate antigen 19-9; AJCC, American Joint Committee on Cancer.

**Table 2 jcm-08-01448-t002:** Univariate and multivariate Cox regression analysis for overall survival.

Variable	Category	Univariate Analysis		Multivariate Analysis	
		HR (95% CI)	*p* Value	Adjusted HR (95% CI)	*p* Value
ECOG PS	0 or 1	1		1	
	2	2.45 (1.72–3.47)	<0.001	2.23 (1.49–3.34)	<0.001
	>3	5.30 (2.29–9.92)	<0.001	3.28 (1.58–6.79)	0.001
CCI	0	1		1	
	1	1.57 (1.14–2.17)	0.006	1.20 (0.85–1.70)	0.31
	≥2	1.87 (1.25–2.79)	0.002	0.88 (0.54–1.43)	0.61
AJCC stage	3A	1		1	
	3B	1.67 (1.09–2.58)	0.020	1.60 (1.02–2.50)	0.040
	3C	3.26 (2.15–4.94)	<0.001	2.36 (1.51–3.70)	<0.001
Body weight loss	<5%	1		1	
	≥5%	2.50 (1.84–3.41)	<0.001	1.75 (1.25–2.47)	0.001
Vascular invasion	No	1		1	
	Yes	1.70 (1.26–2.30)	0.001	1.40 (1.00–1.97)	0.050
Resection margin	Negative	1		1	
	Positive	2.27 (1.50–3.43)	<0.001	1.44 (0.92–2.25)	0.12
Operation method	TG	1		1	
	STG	0.56 (0.44–0.79)	<0.001	1.60 (1.02–2.50)	0.040
Adjuvant chemotherapy	No	1			
	yes	0.47 (0.35–0.64)	<0.001	0.52 (0.38–0.72)	<0.001
GPS	0	1			
	1	2.52 (1.80–3.52)	<0.001	1.97 (1.36–2.86)	<0.001
	2	2.26 (1.45–3.53)	<0.001	1.57 (0.96–2.57)	0.07

ECOG PS, Eastern Cooperative Oncology Group performance status; CCI, Charlson comorbidity index; AJCC, American Joint Committee on Cancer; CEA, carcinoembryonic antigen; CA 19-9, carbohydrate antigen 19-9; TG, total gastrectomy; STG, subtotal gastrectomy; HR, hazard ratio; CI, confidence interval; GPS, Glasgow prognostic score.

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
