# Peer review of "Significance of the Glasgow Prognostic Score in Predicting the Postoperative Outcome of Patients with Stage III Gastric Cancer"

_jcm, 2019, doi:10.3390/jcm8091448_

Round 1

Reviewer 1 Report

Dear Authors,

Thank you for the opportunity to review your submitted manuscript. Overall, I would support this manuscript for publication after minor revisions (see below). Best wishes for the submission.

Minor Comment(s):

In Introduction, statement "Limited studies have used GPS for postoperative prognostication of patients with locally advanced gastric cancer." (line 56) and equally in Discussion, statement "short-term and long-term postoperative outcomes in advanced gastric cancer have not been fully examined." (line 173).

Should at least reference the studies that are available. For example, a cursory literature search finds:

Melling N, Gruning A, Tachezy M, Nentwich M, Reeh M, Uzunoglu FG, et al. Glasgow Prognostic Score may be a prognostic index for overall and perioperative survival in gastric cancer without perioperative treatment. Surgery. 2016;159(6):1548-56.

This study also reports an association between GPS and perioperative mortality. 

Reviewer 2 Report

The paper is well written, however the results are quite obvious: relationship between poorer GPS and survival after surgery is intuitive. I would include much more points to the discussion.

Why did you choose only stage III and not all stages amenable to surgery? What about perioperative chemotherapy? How many patients received pre and post operative CT also according to the latest evidence of FLOT trial? Don't you think that your results are influenced by the chemoterapy administration? A patient with poorer preoperative PS is often not candidated to postoperative treatment. How much this can be a bias to your reported results?

Please improve the two tables: I would split univariate from multivariate and make table 1 easier. 

Round 2

Reviewer 2 Report

I feel that you have addressed all the main points and the present form is acceptable for publication.